# Phage therapy for recurrent urinary tract infections: A qualitative study using the theoretical framework of acceptability

David Wenzel[1]*, Michelle Crick[2], Carolyn Tarrant[3,4], Olivia Revitt[2], Jeni Senior[5], Melissa Haines[6]

1 Population Health Sciences, University of Leicester, United Kingdom, 2 School of Healthcare, Nursing Science, Pregnancy and Child Health Nursing, University of Leicester, Leicester, United Kingdom, 3 School of Medical Sciences, University of Leicester, Leicester, United Kingdom, 4 National Institute for Health and Care Research (NIHR) Greater Manchester Patient Safety Research Collaboration (GM PSRC), Manchester, United Kingdom, 5 School of Healthcare, Nursing Science, Pregnancy and Child Health Nursing, University of Leicester, Leicester, United Kingdom, 6 Becky Mayer Centre for Phage Research, University of Leicester, Leicester, United Kingdom

* DW343@leicester.ac.uk

## Abstract

### Objectives

To explore the anticipated acceptability of bacteriophage therapy for recurrent urinary tract infections (rUTIs) among key stakeholders in the United Kingdom, using the Theoretical Framework of Acceptability (TFA).

### Design

Qualitative study using online focus groups, analysed with reflexive thematic analysis, and mapped to the constructs of the TFA.

### Setting

United Kingdom, conducted online via Microsoft Teams.

### Participants

Forty-four participants across three stakeholder groups: individuals with lived experience of rUTIs, doctors involved in UTI management, and other healthcare professionals (HCPs: specialist nurses, pharmacists, and advanced care practitioners).

### Results

Participants demonstrated high enthusiasm for phage therapy, reflecting positive affective attitude and strong perceptions of effectiveness. For patients with lived experience, enthusiasm was often grounded in desperation following repeated treatment failure, raising ethical concerns around vulnerability to poor-quality research or inequitable access. Across groups, opportunity costs were identified as key barriers,

**Data availability statement:** Data are available upon reasonable request for researchers who meet the criteria for access to confidential data, please contact the study sponsor: rgosponsor@leicester.ac.uk for access. Data is stored securely: R:\Qualitative\mc778\Transcripts.

**Funding:** The study was funded by the Leicester Institute for Advanced Studies (Seed Corn Award). Additional support was provided by the National Institute for Health Research (Academic Clinical Lectureship to MH) and the Wellcome Trust (Doctoral Training Programme in Mental Health and Neurosciences to DW (MHN DTP - 223508/Z/21/Z)). None of the funders had any input into study design or delivery. CT is funded by the NIHR Greater Manchester Patient Safety Research Collaboration (GM PSRC). The views expressed are those of the authors and not necessarily those of the NIHR or the Department of Health and Social Care.

**Competing interests:** The authors have declared that no competing interests exist.

centred on financial impact and availability within the NHS. Gaps in understanding about mechanisms, safety, and delivery highlighted the importance of intervention coherence and, for doctors and other HCPs, self-efficacy. Lived experience participants described strong online communities that acted as informal knowledge networks, suggesting a potential channel for dissemination and education.

## Conclusions

Phage therapy was broadly viewed as acceptable, but acceptability is conditional on evidence of safety and effectiveness, equitable access, and clear communication. Application of the TFA demonstrated how enthusiasm, vulnerability, cost, and coherence intersect to shape overall perceptions. These findings emphasise the importance of ethical safeguards, patient-centred education, and integration with community networks to support the responsible adoption of phage therapy.

## Introduction

Urinary tract infections (UTIs) represent one of the most frequently encountered infections in clinical practice, comprising a spectrum from uncomplicated cystitis to complex presentations such as pyelonephritis and urosepsis [1]. Uncomplicated UTIs affect non-pregnant women without structural or functional urinary tract abnormalities and are extremely common, with almost half of all women experiencing an episode in their lifetime and one in three by the age of 24 [2]. Conversely, UTIs in men are uncommon before later life, with incidence parity emerging only in those aged over 85 years [3]. This gender discrepancy is thought to reflect anatomical differences, including urethral length and proximity of the urethral meatus to the anal canal [4].

A significant proportion of individuals experience recurrent urinary tract infections (rUTIs), defined as two or more infections within six months, or three or more within a year [5]. Almost a third of women experiencing their first episode of UTI will suffer a recurrence within the first six months and almost 3% will suffer two episodes in that period [6]. Recurrence may become persistent and is driven by multiple factors, including host genetics, behavioural practices, and microbial persistence. Recurrent infections often originate from the patient's own uropathogenic flora, with risk heightened following antibiotic-induced depletion of protective vaginal lactobacilli [7].

Beyond these established mechanisms, emerging evidence suggests that rUTIs may also arise from intracellular reservoirs and biofilm formation within the bladder epithelium [8]. These adaptations enable uropathogens to evade immune responses and antimicrobial therapy, resulting in relapsing symptoms and chronic infection. Despite the burden of disease, therapeutic options remain limited. Long-term antibiotic prophylaxis is commonly employed, yet its effectiveness is constrained by side effects and growing concern about antimicrobial resistance.

The impact of rUTI extends beyond physical symptoms. Individuals with recurrent infection report significant psychological distress, anxiety about recurrence, and disruptions to daily life [9,10]. A recent qualitative study of people living with rUTI

highlighted a pattern of marginalisation in healthcare encounters, a sense of therapeutic nihilism, and desperation for more effective and less burdensome treatments [11]. The same study described participants' cautious optimism toward novel interventions, tempered by concerns around safety, efficacy, and access.

## Bacteriophage therapy

The urgent need for alternatives to antibiotic therapy in recurrent urinary tract infections has prompted renewed interest in bacteriophage therapy. Bacteriophages, or phages, are viruses that specifically infect and lyse bacterial cells [12]. First discovered in the early twentieth century, phages were initially explored as therapeutic agents before the antibiotic era, but interest waned following the widespread adoption of antibiotics. In recent years, however, phage therapy has re-emerged as a potential strategy to address antimicrobial resistance and chronic, treatment-refractory infections.

In the context of rUTI, phage therapy offers a mechanism that is both highly specific and potentially less disruptive to the broader human microbiome than conventional antibiotics. Phages are typically selected, or engineered, to target known uropathogens such as *Escherichia coli*, *Klebsiella pneumoniae*, or *Proteus mirabilis*, which are commonly implicated in recurrent infections. Early-phase clinical trials and case series have suggested that phage preparations can reduce bacterial load and improve clinical outcomes, particularly in patients with multidrug-resistant infections or where antibiotic options are limited [13].

Importantly, phage therapy introduces unfamiliar concepts to patients, doctors and allied healthcare professionals (HCPs: specialist nurses, pharmacists, and advanced care practitioners) alike. Its biological nature, delivery mechanisms (which may include oral, topical, or intravesical routes), and potential need for personalised matching to specific pathogens distinguish it from traditional treatment paradigms. This novelty may influence perceptions of safety, efficacy, and overall acceptability.

## Theoretical framework

As interest in phage therapy and other non-antibiotic approaches grows, it becomes increasingly important to understand how these interventions are perceived by patients and other stakeholders. Novel treatments, particularly those that depart significantly from conventional treatment paradigms, require not only evidence of efficacy and safety but also consideration of how they are understood, valued, and accepted by those offered them. Acceptability has emerged as a critical construct in implementation research and is increasingly recognised as a determinant of engagement, adherence, and implementation success [14].

The Theoretical Framework of Acceptability (TFA) provides a structured approach to conceptualising and evaluating the acceptability of healthcare interventions. Developed through a synthesis of empirical literature and theoretical reasoning, the TFA defines acceptability as "a multi-faceted construct that reflects the extent to which people delivering or receiving a healthcare intervention consider it to be appropriate, based on anticipated or experienced cognitive and emotional responses to the intervention" (Sekhon et al., 2017, p. 4).

The TFA comprises seven constructs: affective attitude, burden, ethicality, intervention coherence, opportunity costs, perceived effectiveness, and self-efficacy. The TFA domains are summarised in Table 1.

Crucially, Sekhon and colleagues emphasise that the timing of assessment, whether before, during, or after treatment exposure, can profoundly shape how interventions are appraised. Clarity around participants' experience in relation to the intervention is therefore essential when applying the framework.

In this study, we use the TFA to explore how individuals with rUTI, and doctors and other healthcare professionals with a role in caring for people with rUTIs, cognitively and emotionally appraise phage therapy as a novel treatment. By analysing responses across the TFA's domains, this research aims to inform not only the future clinical rollout of such therapies but also how they are introduced and communicated to patients and healthcare staff in a way that supports trust, understanding, and uptake.

**Table 1. Domains within the theoretical framework of acceptability.**

| Domain | Description |
| --- | --- |
| **Affective Attitude** | Emotional response to the intervention. |
| **Burden** | Perceived effort required to engage with the intervention. |
| **Perceived Effectiveness** | Belief in the intervention's ability to achieve its intended outcomes. |
| **Ethicality** | Alignment of the intervention with personal values and ethical standards. |
| **Intervention Coherence** | Understanding of how the intervention works and its relevance. |
| **Opportunity Costs** | Perceived trade-offs, such as time or resources, required to participate. |
| **Self-Efficacy** | Confidence in one's ability to engage with the intervention. |

TFA Descriptors.

## Methods

### Aims

The aim of this study was to explore stakeholder perceptions of acceptability of bacteriophage therapy as a potential treatment for UTIs in the UK, using the TFA. This work forms part of preparatory research for the possible integration of phage therapy into National Health Service care pathways, and future phage research.

### Recruitment

Our patient and public input group identified three key stakeholder groups: doctors who treat UTIs, other healthcare professionals (specialist nurses, pharmacists and advanced care practitioners) and individuals with lived experience of rUTIs. Initial recruitment of these three stakeholder groups was undertaken via promotional materials disseminated through the University of Leicester media channels, supported by Antibiotic Research UK (recently renamed AMR Action UK) and Bladder Health UK. However, due to slower than expected recruitment among individuals with lived experience, additional approaches to recruitment were implemented. Advertising materials were shared via the researchers' social media (Twitter and LinkedIn), which generated over 100 expressions of interest within 48 hours. From these respondents, participants were randomly selected to ensure diversity in age, gender, and background whilst ensuring UK residence in order to address the study aims. Active recruitment of participants took place between 19 April 2024–7 June 2024.

### Focus groups

Focus groups were selected over individual interviews as they allowed participants to collectively make sense of the novel concept of phage therapy, supporting exploration of intervention coherence within the TFA. The interactive format enabled participants to question, clarify, and build upon each other's understanding, which was particularly valuable given the unfamiliarity of this therapeutic approach. In addition, for people with rUTI who frequently report experiences of marginalisation in healthcare encounters the group setting offered a supportive environment in which participants could validate and normalise one another's perspectives.

Online delivery, via Microsoft Teams, was chosen to maximise accessibility and geographical reach, with recognition of the trade-off in capturing non-verbal communication [15]. Each session was hosted by two facilitators: one discussion lead and one presenter. The sessions commenced with a presentation and video explaining phage therapy in lay terms (https://youtu.be/7YBjsBRsVfo), to ensure that all participants had been introduced to phage therapy prior to the focus group discussion. A copy of the topic guide is shown in S1 Fig. All sessions were recorded and transcribed using Microsoft Teams' built-in functions. Transcripts were manually reviewed for accuracy by author MC and pseudonymised during this process. Audio recordings were securely stored and deleted after verification.

## Data analysis

The study was informed by the Theoretical Framework of Acceptability (TFA), which provided a sensitising framework for exploring anticipated acceptability. A reflexive thematic analysis (RFA) approach was used to analyse data, following the six-step process described by Braun and Clarke [16]. All transcripts were reviewed in full by the research team for orientation to the dataset.

Initial coding was conducted inductively, with codes generated directly from the data without applying TFA constructs as an a priori coding template. Coding was undertaken within stakeholder groups (lived experience, doctors, healthcare professionals) before patterns were examined across groups.

After completion of the first four focus groups (two lived experience and two healthcare professional groups), the research team met to review early analytic findings and assess information power, guided by Malterud, Serisma and Guassora [17]. At this stage, we also considered how developing themes related to the TFA constructs, as the pre-identified analytic framework. This process involved examining areas of alignment, divergence, and potential extension, without restructuring the coding framework deductively.

Two further focus groups with doctors were subsequently conducted to ensure representation across stakeholder categories. Following analysis of all six focus groups, the research team reconvened to reassess information power and agreed by consensus that sufficient depth and variation had been achieved to address the study aims.

Following development of inductive codes and preliminary themes, a secondary interpretive step involved mapping themes onto the component constructs of the TFA. This mapping was collaborative and theory-informed. Themes that did not align with existing TFA constructs were explicitly retained and described. Data management and coding were supported using NVivo software.

To enhance analytic rigour, the research team reviewed the final analysis against Braun and Clarke's 20-question framework for quality in reflexive thematic analysis [18]. Consistent with the philosophical and epistemological underpinnings of this approach, no quantitative analysis of the data was planned or undertaken.

## Ethics

All participants completed written consent via an electronic consent form following an opportunity to review all study materials. Ethical scrutiny for the study was given by the University of Leicester Ethics Committee (reference: 24411-mh508-ls:genetics&genomebiology).

## Reflexivity

In keeping with a reflexive thematic analysis approach, the research team remained attentive to how their disciplinary backgrounds and prior engagement with qualitative research and phage-related scholarship may have shaped data interpretation throughout analysis. The research team comprised of doctors, HCPs and health researchers with backgrounds spanning acute medical care, allied health, qualitative research, and bacteriophage science. Reflexive discussions were used throughout analysis to consider how differing levels of clinical experience with recurrent urinary tract infection and prior engagement with phage research might shape interpretation.

## Findings

Six online focus groups were conducted, two for each stakeholder group, via Microsoft Teams, each lasting 90 minutes – demographic details are listed in appendix B.

Analysis of focus group discussions revealed a range of perspectives on bacteriophage therapy, shaped by participants' professional roles, clinical experience, and personal encounters with recurrent urinary tract infections. Mapping

these perspectives to the constructs of the Theoretical Framework of Acceptability (TFA) highlighted both shared and stakeholder-specific considerations regarding the potential adoption of phage therapy. We identified three cross cutting themes which incorporated constructs from the TFA: Perceptions of phage therapy and its value; Practicalities of implementation and access; and understanding and coherence.

## Theme 1: Perceptions of phage therapy and its value

(*Affective Attitude, Perceived Effectiveness, Ethicality*)

Stakeholders' perceptions of phage therapy centred on its potential value, the hope it offered, and the evidence supporting its effectiveness. Across all stakeholder groups, views were shaped by prior knowledge, personal or professional experience, and the influence of peers and colleagues.

For people with lived experience of rUTIs, perceptions of phage therapy were driven by a reaction to dwindling treatment options. Phage therapy was primarily viewed as a last-resort option, providing hope where conventional treatments had failed. Some patients with AMR-related infections described their situation as life-threatening, framing access to phage as essential to survival. Urgency and personal stakes on the whole outweighed concerns about the novelty of the treatment, or limited clinical evidence:

*"I am literally on dwindling options now. I have been told that by the doctor"*

Lived Experience Group 2, Participant RW

*"Desperation would make me want to try it"*

Lived Experience Group 1, Participant DN

Feelings of frustration and inequity were common among patients who were aware of phage availability abroad but found it inaccessible in the UK.

Doctors and other healthcare professionals tended to be more cautious about the potential for phage therapy. They primarily emphasised the need for robust clinical evidence from controlled trials, supported by clear data on treatment success rates and survival outcomes, as prerequisites for trust and uptake.

For these stakeholders, the perceived value of phage therapy was closely tied to evidence of effectiveness and safety. In the absence of such data, stakeholders anticipated hesitancy in recommending or accepting phage therapy, regardless of its theoretical potential. Many participants reported having encountered accounts of phage success, often originating from countries outside the UK. While these reports generated interest, they were not wholly considered a substitute for locally generated, high-quality evidence.

Aligned with the views of patients, health professionals did, however, argue for the acceptability of using phage therapy where it would be the last resort, and could impact on survival. In such cases professionals also felt that the severity and urgency of a patient's situation could reduce the perceived need for absolute proof of effectiveness – some participants drew parallels to other high stakes situations where the risk of harm is balanced against clinical efficacy.

*"If a patient is willing to do chemo for cancer, that kills good and bad cells, then why not phage therapy? That there is a minimal risk factors or adverse effects. There is a lot of adverse effects on antibiotics."* Healthcare Professionals group 1, participant ZD

Across groups, phage was regarded as adding value to the field of urology and the management of rUTIs. Patients, in particular, tended to view it as not only potentially effective but also worth the effort, cost, and potential risks involved.

*"I would seek for therapy that gives me the long-term result, even if it means that I need to wait quite a while to see the results"*

Lived Experience group 1, participant LW

*"I've had sepsis. So I'm more than happy to have this therapy if it becomes available to try it"*

Lived Experience 2, Participant KS

Ultimately, for many participants across both stakeholder groups, phage represented a source of hope for patients with rUTIs, in the face of limited alternatives. Yet this very desperation was seen as reinforcing the need for strong ethical safeguards. Participants highlighted the need for shared decision-making about phage therapy use in individual cases, underpinned by sufficient education and awareness among healthcare providers and patients, to ensure that phage therapy was used in ethically-appropriate ways.

*"We don't know about the cost. We don't know about the side effects… they've done a couple of clinical trials on people, but we don't know the long term effects."*

Healthcare Professionals group 1, participant GR

They emphasised the importance of healthcare professionals listening to patients, and giving adequate time and attention to supporting truly informed consent.

Phage therapy was also invested with significant value and hope in the broader context of antimicrobial resistance. Its distinctive mechanism of action was viewed as a significant advancement, offering hope to individual patients, but also, hope for future treatment options, reducing reliance on the depleted resource of antibiotics in the context of the emergence and spread of AMR. Concerns about possible side effects and unknown long-term consequences were balanced against perceived gains, including the potential to address the growing threat of antimicrobial resistance (AMR).

*"It's just weird that people don't really realise about the pandemic of antibiotic resistance and that is going to kill is it 30 million or something by 2050"*

Lived Experience group 2, Participant RW

*"Some patients have gotten to the point where I mean, nothing is working for them. And back to your antibiotic drugs that should have worked for them, don't work more because of the increase in resistance to this antibacterial. So, it's a good development to have something work other than the normal antibacterial that we that administer to them."*

Lived Experience group 1, participant FG

## Theme 2: Practicalities of implementation and access

*(Burden, Opportunity costs, Safety & Risk, Self-Efficacy)*

Participants highlighted uncertainties about how phage therapy would be delivered, monitored, and integrated into existing pathways. Most doctors felt that prescribing, administration, and monitoring should be led by specialist services, typically in a hospital setting, with elements of follow-up potentially delivered in primary care.

*"When it comes to prescribing phage therapy, the responsibility should be with the infectious disease specialist and the urologist"*

Doctors group 2, participant BJ

Those with lived experience of rUTIs emphasised the need for ease of access within clear, consistently applied care pathways. This reflected their complex treatment journeys and a perception of inconsistency in how rUTIs were managed across services.

*"Would it vary between GP practices? Some are giving out antibiotics like sweets and others are with-holding them"*

Lived Experience group 1, participant DN

Financial aspects were discussed across groups, with concerns that affordability could shape availability. Participants questioned whether phage therapy would be funded by the NHS or require private payment, expressing concern that cost could limit access, particularly for those unable to self-fund. These concerns extended to the potential burden on health-care budgets, which some feared might restrict availability even when phage was clinically indicated.

*"The main barrier will always be the financial aspect and awareness."*

Doctors group 2, participant JH

Safety was a persistent cross-cutting issue. Stakeholders expressed anxiety about unknown risks for different population groups, which would need to be understood to allow implementation – particularly for vulnerable populations such as children or during pregnancy.

*"Pregnant women, for example. Some of the antibiotics we have, handful are contra-indicated in pregnancy and so [will] phage therapy be a better option or will that have side effects as well? […] While breastfeeding, is it advisable?"*

Healthcare Professionals group 1, participant DW

For patients with rUTIs who felt they had exhausted all other treatment options, willingness to accept phage therapy as a last resort could come with the burden of concerns about the safety of an unfamiliar treatment. The weight of making such a decision under these circumstances was seen as an important, and sometimes overlooked, aspect of the treatment's acceptability. Participants emphasised the vital need for research, outcomes data, and education, among all stakeholders to build confidence in the treatment's safety profile.

*"There's going to be a lot of concern and scepticism. A lot of people are going to be worried about the safety questions, and have concerns about the safety and potential side effects"*

Doctors group 2, participant BJ

Participants from the lived experienced group often described using social media platforms to explore information related to new treatments, including phage.

*"Through all the Facebook pages that I've been on when you do get the odd comment from somebody who's had [bacteriophage therapy], it's either been that it's helped or it hasn't worked."*

Lived Experience group 1, participant CW

Some participants raised concerns about the burden of administration, especially if phage requires self-administered injections.

*"I would say that if it was an injection it, whether if I've got to have it regularly, they might get me to inject myself and it which case I don't think I could do it."*

Lived Experience group 1, Participant DN

For healthcare professionals, self-efficacy related to their confidence in delivering phage therapy within their professional scope of practice. Participants frequently emphasised the need for targeted training and ongoing support to equip HCPs with the knowledge and skills required to make informed prescribing decisions or oversee phage therapy.

**Theme 3: Understanding and coherence**

(*Intervention Coherence*)

Although most participants had heard of phage, unsurprisingly given their interest in rUTI treatment, it was still regarded as a novel treatment, with participants tending to report limited knowledge and understanding. Understanding of the therapy and how it worked was seen as vital for acceptability.

*"I don't have so much knowledge about the phage therapy, so I'm not aware of the side effects. I'm not aware of contraindications. And also, method of usage […] What are the side effects on the long run? What do we think? These bacteria, will they get resistance to the bacteriophages?"*

Lived Experience group 1, Participant DW

The principle of using viruses to treat bacterial infections provoked mixed reactions. Patients and professionals alike voiced questions about how phages worked, whether they might mutate, what happened once infections were cleared, how long treatment should last, and whether the treatment could be reversed. A lack of understanding of practical aspects such as appropriate dosage, likely duration of treatment, and potential contraindications was also evident. Additionally, the COVID-19 pandemic had heightened public awareness of viruses and vaccines, creating in some cases, lingering scepticism or fear.

*"There's going to be lots of questions about how phage therapy is administered and what happens to the bacterio-phages after the infection is cleared"*

Doctors group 2, participant BJ

*"After bacteria has been eliminated, what happens to the virus? Does it remain in the patient's system? Does it affect the patient? We should know before administering as we need to get consent."*

Healthcare Professionals group 1, participant DJ

*"One of the issues people had with COVID vaccine was the fact that it was a viral vaccine and people were scared of the fact that there could be mutation in the long term. You know, as a long-term effect of this vaccine. So that was what was really affecting people's ability to take […] the vaccine when it was out there. So I think that might be a problem to the patients that have to take this phage therapy."*

Healthcare Professionals group 1, participant GR

These knowledge gaps were seen as potential barriers to acceptance, underscoring the need for clear, accessible education for both patients and healthcare professionals.

Again, phage therapy was weighed against antibiotic therapy, in terms of familiarity and coherence: antibiotics were perceived as a familiar, cheap, convenient, and preferred treatment.

*"Definitely people may decide to go for the antibiotic"*

Doctors group, 2, participant JH

This familiarity underpinned the views of many professionals that their preference in the first instance would be to try alternative forms of antibiotics where patients had resistant infections, as opposed to recommending phage therapy.

*"I don't think I will advise someone to go further on it, rather than using the other formats of antibiotics, which seems to be more convenient to people"*

Healthcare Professionals group 1, participant CD

These gaps in intervention coherence reinforced the importance of evidence and information as a foundation for acceptability.

*"There should be some kind of orientation when it comes to phage therapy, so as to make everyone understand, […] 'cause I feel many people wouldn't want to engage in something they don't feel comfortable with, something they don't understand. So when there's basic knowledge on what phage therapy can and cannot do, or it can produce or what it cannot, it brings more people to understanding and more people feel [they can] use phage therapy."*

Healthcare Professionals group 1, participant CW

## Discussion

This study represents one of the first applications of the Theoretical Framework of Acceptability (TFA) to bacteriophage therapy. The framework has been employed across a variety of healthcare settings, from surgical interventions [19] to health programmes [20], but rarely in relation to therapies that remain largely experimental. Applying the TFA here allowed us to examine how phage therapy is cognitively and emotionally appraised by both patients and healthcare professionals at a time when clinical experience of the intervention is still evolving.

Overall, participants demonstrated high levels of enthusiasm for phage therapy, with many emphasising its potential to provide hope where conventional antibiotic regimens had failed. Within the TFA, this reflects a positive affective attitude and strong perceptions of perceived effectiveness. However, enthusiasm was often grounded in desperation, particularly among participants with lived experience of recurrent UTIs who reported exhausting existing options. Work with US populations of physicians echo these findings, where concern around AMR becomes a driving factor in the acceptability of phage [21].

This dynamic raises an important ethical concern. Desperation may heighten willingness to accept unproven interventions, potentially increasing vulnerability to poor-quality research or inequitable implementation [22,23]. Situating these findings within ethicality, the results highlight the importance of rigorous safeguards to ensure that patient hope is not exploited in ways that compromise autonomy or safety. Comparable issues have been described in oncology, where patients participating in early-phase trials rarely derive clinical benefit, yet their desperation to access treatment powerfully shapes consent [24]. These patterns resonate with the concept of therapeutic misconception, in which patients conflate participation in research with guaranteed therapeutic benefit, underscoring the importance of careful consent processes in phage research.

 

Analysis of the data suggests the greatest barriers to acceptability are opportunity costs and these were specifically focused around the financial impact and ease of access to phage. Financial cost often determines accessibility, linking these two constructs. Concerns expressed by participants are well founded, as previous examples of restricted prescribing practice have demonstrated that high-cost antimicrobial therapies can be rationed or delayed in the NHS [25]. Such constraints not only affect equity of access but may also erode the positive affective attitude patients initially hold towards novel therapies.

Limited awareness and understanding of phage therapy emerged as another recurring theme. Patients and professionals alike reported uncertainties about mechanisms of action, long-term safety, and practical delivery. These gaps undermine intervention coherence and, for doctors and HCPs, self-efficacy. Participants repeatedly expressed the need for targeted education and communication strategies to support both groups in making informed decisions. For patients, clarity about what phage can and cannot do was viewed as central to informed consent. For healthcare professionals, access to robust data and tailored training were seen as prerequisites for confidence in prescribing or delivering phage therapy.

A distinctive finding was the extent to which lived experience participants drew on online peer networks to inform their understanding. These groups fulfil the criteria of being a community through linked social ties that formed around shared common perspectives [26], specifically around suffering from rUTIs. Geographically this community exists online, an emerging characteristic of modern community groups [27]. This community shares information, studies, opinions and values away from medical or scientific scrutiny. Accessing and leveraging these information distribution channels may facilitate improving intervention coherence and hence, acceptability. This finding emerged inductively during analysis rather than from a priori application of the TFA; as such, it is interpretive in nature and sits partially outside the framework, warranting further exploration.

Taken together, these findings illustrate both the promise and the challenges of introducing phage therapy for rUTIs. Acceptability is high, but it is also conditional: enthusiasm is tempered by concerns about safety, coherence, opportunity costs, and equity of access. By applying the TFA, we were able to surface not only practical barriers but also ethical considerations around vulnerability and informed consent. More broadly, this study demonstrates the utility of the TFA in evaluating acceptability for novel and experimental therapies, highlighting the value of prospective assessments that anticipate implementation challenges before widespread clinical rollout.

## Limitations

This study has several limitations that should be considered when interpreting the findings. First, phage therapy itself remains under development, with uncertainty around how products will ultimately be manufactured, regulated, and delivered within health systems. While participants were able to engage with the general concept of bacteriophages as an alternative to antibiotics, their views were necessarily hypothetical. As a result, the findings reflect broad perceptions of acceptability rather than responses to a fully defined treatment pathway.

Second, this study captures *anticipated* rather than *experienced* acceptability. Focus group participants were asked to appraise a novel therapy they had not personally received. Although this prospective perspective is valuable for informing early implementation strategies, acceptability may shift once patients, doctors or HCPs are directly exposed to treatment. Longitudinal assessment, including follow-up during or after clinical use, will be important to build on these findings.

Third, our sample included self-selecting participants with lived experience of rUTI. We did not undertake clinical validation of participants' diagnostic history or medical background, which means findings are based on self-reported experience. While this reflects the epistemological stance of valuing lived experience, it may limit certainty about the clinical comparability of the sample.

This study employed reflexive thematic analysis to explore anticipated acceptability in depth rather than to quantify differences between stakeholder groups. While the findings may inform future quantitative or mixed-methods research examining acceptability across TFA constructs [28], such analyses were beyond the scope of this paper.

Finally, recruitment was supported through phage researchers' social media networks and partner charities. This approach enabled rapid recruitment but may have attracted individuals with pre-existing interest in novel therapies or antimicrobial resistance research. These participants may have been more motivated, knowledgeable, or enthusiastic about phage therapy than the wider population of people with rUTI or doctors and HCPs in routine practice. This may have influenced the prominence of themes relating to hope, innovation, and willingness to consider experimental therapies. While this is consistent with the study aim of exploring anticipated acceptability among engaged stakeholders, it limits transferability to less research-engaged or more sceptical populations. Findings should therefore be interpreted as reflecting the perspectives of a motivated and information-seeking sample rather than a representative cross-section of all stakeholders.

## Conclusion

Phage therapy was viewed by patients and healthcare professionals as a promising and acceptable option for recurrent urinary tract infections, particularly where conventional treatments had failed. However, this enthusiasm was often shaped by desperation, highlighting the need for safeguards against premature or inequitable implementation. Using the Theoretical Framework of Acceptability allowed us to identify how perceptions of effectiveness, burden, coherence, and opportunity costs intersect to shape overall acceptability. Clear communication, equitable access, and careful consent processes will be essential if phage therapy is to be responsibly integrated into future clinical care.

## Supporting information

**S1 Fig. Topic guide.**
(PDF)

**S2 Fig. Table 1: Date and times of focus groups, stakeholder group, number of participants and length where recorded.** Table 2: Number of participants per stakeholder group, country of origin and gender.
(DOCX)

## Acknowledgments

We are grateful to the healthcare staff and people with rUTIs who gave up their time to participate in focus groups. The authors used OpenAI's ChatGPT (GPT-5) to assist in refining readability and language. All content was reviewed, edited, and approved by the authors, who take full responsibility for the final text.

## Author contributions

**Conceptualization:** Michelle Crick, Melissa Haines.

**Data curation:** David Wenzel, Michelle Crick.

**Formal analysis:** David Wenzel, Michelle Crick.

**Funding acquisition:** David Wenzel, Melissa Haines.

**Investigation:** Carolyn Tarrant, Olivia Revitt, Jeni Senior.

**Methodology:** Carolyn Tarrant, Olivia Revitt, Jeni Senior.

**Project administration:** Michelle Crick, Melissa Haines.

**Supervision:** Carolyn Tarrant, Melissa Haines.

**Writing – original draft:** David Wenzel.

**Writing – review & editing:** David Wenzel, Michelle Crick, Carolyn Tarrant, Olivia Revitt, Jeni Senior, Melissa Haines.

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
