## [Decision Letter · Decision Letter 0]

19 Jan 2026

PONE-D-25-53420Phage Therapy for Recurrent Urinary Tract Infections: A Qualitative Study Using the Theoretical Framework of AcceptabilityPLOS One

Dear Dr. Wenzel,

Thank you for submitting your manuscript to PLOS ONE. After careful consideration, we feel that it has merit but does not fully meet PLOS ONE’s publication criteria as it currently stands. Therefore, we invite you to submit a revised version of the manuscript that addresses the points raised during the review process.

We look forward to receiving your revised manuscript.

Kind regards,

Awatif Abid Al-Judaibi, PhD

Academic Editor

PLOS One

Journal Requirements:

3. Please expand the acronym “NIHR” (as indicated in your financial disclosure) so that it states the name of your funders in full.

4. In the online submission form you indicate that your data is not available for proprietary reasons and have provided a contact point for accessing this data. Please note that your current contact point is a co-author on this manuscript. According to our Data Policy, the contact point must not be an author on the manuscript and must be an institutional contact, ideally not an individual. Please revise your data statement to a non-author institutional point of contact, such as a data access or ethics committee, and send this to us via return email. Please also include contact information for the third party organization, and please include the full citation of where the data can be found.

6. Please remove all personal information/transcripts within your manuscript, ensure that the data shared are in accordance with participant consent, and re-upload a fully anonymized data set.

Reviewers' comments:

Reviewer's Responses to Questions

**Comments to the Author**

1. Is the manuscript technically sound, and do the data support the conclusions?

Reviewer #1: Yes

Reviewer #2: Partly

2. Has the statistical analysis been performed appropriately and rigorously? 

Reviewer #1: N/A

Reviewer #2: N/A

3. Have the authors made all data underlying the findings in their manuscript fully available?

Reviewer #1: Yes

Reviewer #2: Yes

4. Is the manuscript presented in an intelligible fashion and written in standard English?

Reviewer #1: Yes

Reviewer #2: Yes

5. Review Comments to the Author

Reviewer #1: This manuscript presents a rigorous, well-executed qualitative study exploring the anticipated acceptability of bacteriophage therapy for recurrent urinary tract infections using the Theoretical Framework of Acceptability. The topic is timely, ethically important, and well suited to PLOS ONE’s scope. The manuscript is clearly written, methodologically sound, and demonstrates strong engagement with both theory and empirical data.

Minor points for improvement

1. Reflexivity

While reflexive thematic analysis is cited, a brief reflexivity statement (e.g., researchers’ disciplinary backgrounds or relationship to phage research) would further strengthen methodological transparency.

2. Online community findings

The discussion of online peer communities is insightful. The authors may wish to clarify more explicitly that this theme emerged inductively and sits partially outside the TFA, as they note, to avoid any perception of conceptual overextension.

3. Terminology consistency

Minor improvements could be made by ensuring consistent terminology when referring to stakeholder groups (e.g., “doctors” vs. “clinicians”) throughout the manuscript.

Reviewer #2: Good, overall. This manuscript does a nice job of introducing the TFA model and how it relates to phage therapy here. The discussion section aptly summarizes the anecdotal statements made by the 6 online focus groups. It is clear that desperate rUTI patients are most likely to view phage therapy as an appropriate treatment; it is also evident that financial cost and the lack of knowledge surrounding phage therapy are clear deterrents. I also appreciate that the study's limitations were elaborated upon. Furthermore, the entire manuscript is well-written.

However, I am a bit concerned about the lack of quantitative analyses done here, with respect to affective attitude, perceived effectiveness, ethicality, burden, opportunity costs, etc. Is there a way to quantify the responses of the 3 stakeholder groups and how they differ across the 7 constructs of the TFA? That way, the reader would be more readily able to determine how, for instance, doctors and healthcare professionals (i.e.-2 of the 3 stakeholder groups detailed here) differ in their viewpoints of phage therapy's burden, opportunity costs, ethics, etc. Although anecdotal statements from the focus groups, as done here, are somewhat insightful, quantitative analyses would prove to be even more revealing. For example, the authors could parse through the focus group discussions and quantify the frequency of statements associated with "positive affection" or "negative affection" for the "Affective Attitude" arm of the TFA. Alternatively, if the authors asked the respondents to rank perceived effectiveness on a quantitative scale, this would allow us to understand how the 3 different stakeholder groups differ on the basis of concrete numerical mean differences (e.g.-a "perceived effectiveness" mean score for each group). All in all, there are many ways that the authors could add a quantitative dimension to this study, addressing the 7 constructs of the TFA across the 3 stakeholder groups. This would be greatly appreciated.

6. PLOS authors have the option to publish the peer review history of their article (what does this mean?). If published, this will include your full peer review and any attached files.

Reviewer #1: **Yes:** Ali Bin Thani

Reviewer #2: No

---

## [Author Response · Author response to Decision Letter 1]

5 Feb 2026

Comments are replied to individually in the 'response to reviewers' file.

---

## [Decision Letter · Decision Letter 1]

3 Mar 2026

PONE-D-25-53420R1Phage Therapy for Recurrent Urinary Tract Infections: A Qualitative Study Using the Theoretical Framework of AcceptabilityPLOS One

Dear Dr. Wenzel,

Thank you for submitting your manuscript to PLOS ONE. After careful consideration, we feel that it has merit but does not fully meet PLOS ONE’s publication criteria as it currently stands. Therefore, we invite you to submit a revised version of the manuscript that addresses the points raised during the review process.

We look forward to receiving your revised manuscript.

Kind regards,

Awatif Abid Al-Judaibi, PhD

Academic Editor

PLOS One

Journal Requirements:

Reviewers' comments:

Reviewer's Responses to Questions

**Comments to the Author**

1. If the authors have adequately addressed your comments raised in a previous round of review and you feel that this manuscript is now acceptable for publication, you may indicate that here to bypass the “Comments to the Author” section, enter your conflict of interest statement in the “Confidential to Editor” section, and submit your "Accept" recommendation.

Reviewer #2: All comments have been addressed

Reviewer #3: All comments have been addressed

Reviewer #4: (No Response)

2. Is the manuscript technically sound, and do the data support the conclusions?

Reviewer #2: (No Response)

Reviewer #3: Yes

Reviewer #4: Partly

3. Has the statistical analysis been performed appropriately and rigorously? 

Reviewer #2: (No Response)

Reviewer #3: Yes

Reviewer #4: Yes

4. Have the authors made all data underlying the findings in their manuscript fully available?

Reviewer #2: (No Response)

Reviewer #3: Yes

Reviewer #4: Yes

5. Is the manuscript presented in an intelligible fashion and written in standard English?

Reviewer #2: (No Response)

Reviewer #3: Yes

Reviewer #4: Yes

6. Review Comments to the Author

Reviewer #2: (No Response)

Reviewer #3: (No Response)

Reviewer #4: This manuscript presents a qualitative study exploring stakeholder perceptions regarding the anticipated acceptability of bacteriophage therapy for recurrent urinary tract infections using the Theoretical Framework of Acceptability. The qualitative design and use of focus groups are appropriate for the stated aims, and the conclusions are generally supported by the data presented. The manuscript is clearly structured overall; however, several issues should be addressed to improve methodological transparency and reporting quality.

First, recruitment through social media and networks connected to phage research may have introduced selection bias toward participants already interested in novel therapies. The implications of this for transferability of findings should be more explicitly discussed. In addition, further clarification is needed regarding how “information power” was determined and how inductive coding was subsequently mapped onto TFA constructs, as this process is currently insufficiently described.

Second, the Data Availability Statement indicates that data are available only upon reasonable request. This does not appear fully aligned with PLOS ONE data availability requirements unless justified ethical restrictions apply. The authors should clarify whether anonymized qualitative data can be deposited in a repository or provide a clearer explanation of access procedures and restrictions.

Several presentation issues also require attention. Some references cited in the Introduction (e.g., early epidemiological statements) appear to rely on secondary sources rather than original studies and should be revised where appropriate. Referencing style is inconsistent throughout the manuscript and should be standardized according to journal guidelines. Tables currently lack clear descriptive titles and should be revised so they can be interpreted independently of the main text. Finally, formatting artifacts and editing marks visible in the manuscript should be removed prior to publication.

7. PLOS authors have the option to publish the peer review history of their article (what does this mean?). If published, this will include your full peer review and any attached files.

Reviewer #2: No

Reviewer #3: No

Reviewer #4: **Yes:** Zaid Iyad Alkhatib

---

## [Author Response · Author response to Decision Letter 2]

3 Mar 2026

Comments to the Author

1. If the authors have adequately addressed your comments raised in a previous round of review and you feel that this manuscript is now acceptable for publication, you may indicate that here to bypass the “Comments to the Author” section, enter your conflict of interest statement in the “Confidential to Editor” section, and submit your "Accept" recommendation.

Reviewer #2: All comments have been addressed

Reviewer #3: All comments have been addressed

Reviewer #4: (No Response)

2. Is the manuscript technically sound, and do the data support the conclusions?

Reviewer #2: (No Response)

Reviewer #3: Yes

Reviewer #4: Partly

3. Has the statistical analysis been performed appropriately and rigorously?

Reviewer #2: (No Response)

Reviewer #3: Yes

Reviewer #4: Yes

4. Have the authors made all data underlying the findings in their manuscript fully available?

Reviewer #2: (No Response)

Reviewer #3: Yes

Reviewer #4: Yes

5. Is the manuscript presented in an intelligible fashion and written in standard English?

Reviewer #2: (No Response)

Reviewer #3: Yes

Reviewer #4: Yes

6. Review Comments to the Author

Reviewer #2: (No Response)

Reviewer #3: (No Response)

Reviewer #4: This manuscript presents a qualitative study exploring stakeholder perceptions regarding the anticipated acceptability of bacteriophage therapy for recurrent urinary tract infections using the Theoretical Framework of Acceptability. The qualitative design and use of focus groups are appropriate for the stated aims, and the conclusions are generally supported by the data presented. The manuscript is clearly structured overall; however, several issues should be addressed to improve methodological transparency and reporting quality.

First, recruitment through social media and networks connected to phage research may have introduced selection bias toward participants already interested in novel therapies. The implications of this for transferability of findings should be more explicitly discussed.

We thank the reviewer for this comment. The Limitations section already acknowledged the implications of recruitment through phage-related networks for transferability. We have now revised this paragraph to make this discussion more explicit and to provide additional analytic depth.

In addition, further clarification is needed regarding how “information power” was determined…

We thank the reviewer for this helpful comment. We agree that greater methodological transparency was warranted. Unfortunately, some of this more granular detail was removed for length, but we are glad to now be able to include it.

We have revised the Methods section to provide a clearer account of how information power was assessed and how the analytic process proceeded. Specifically, we now clarify that information power was reviewed iteratively during data collection, guided by Malterud et al.’s paper. We had two study committee meetings regarding information power, after the first 4 focus groups and after all 6 had been completed. In these meetings we assessed the components of information power that were not known a priori including specificity, dialogue and alignment with the theoretical framework.

… and how inductive coding was subsequently mapped onto TFA constructs, as this process is currently insufficiently described.

We have additionally addressed the transparency of our coding process. The Data Analysis section has been re-written in full to create a more coherent, easy to follow process. Rather than addressing analysis first and information power second the two constructs are addressed alongside one another to plot better against the data analysis meetings that were held. We believe this makes the process easier to follow whilst increasing transparency.

Second, the Data Availability Statement indicates that data are available only upon reasonable request. This does not appear fully aligned with PLOS ONE data availability requirements unless justified ethical restrictions apply. The authors should clarify whether anonymized qualitative data can be deposited in a repository or provide a clearer explanation of access procedures and restrictions.

We thank the reviewer for this important point.

The qualitative dataset contains detailed personal and professional accounts relating to clinical practice and lived health experiences. Although transcripts were anonymised, the depth and contextual specificity of the discussions mean that full public deposition would carry a risk of deductive disclosure. Ethical approval for the study was granted on the basis that data would be stored securely and accessed under controlled conditions rather than deposited in an open repository.

We have discussed the Data Availability Statement with the journal office during the previous revision round, and the current approach was confirmed to be consistent with PLOS ONE policy for qualitative research involving sensitive data. We have now revised the wording to provide a clearer explanation of the access procedures and restrictions, including institutional contact details and governance requirements for researchers seeking access to additional anonymised material.

We hope this clarification addresses the reviewer’s concern.

Several presentation issues also require attention. Some references cited in the Introduction (e.g., early epidemiological statements) appear to rely on secondary sources rather than original studies and should be revised where appropriate. Referencing style is inconsistent throughout the manuscript and should be standardized according to journal guidelines. Tables currently lack clear descriptive titles and should be revised so they can be interpreted independently of the main text. Finally, formatting artifacts and editing marks visible in the manuscript should be removed prior to publication.

We thank the reviewer for these helpful observations.

We have revised the Introduction to ensure that early epidemiological statements are supported by appropriate primary sources, including replacing the citation of Germanos et al. with Schaeffer et al. where this more directly reflects the original evidence.

The reference list has been carefully reviewed and standardised in accordance with PLOS ONE guidance, using Vancouver style consistently throughout. Formatting inconsistencies introduced during manuscript preparation have now been corrected to ensure uniform presentation.

Table titles have been revised to provide clearer, more descriptive headings so that each table can be interpreted independently of the main text.

The manuscript has also been reviewed in full to remove formatting artefacts and ensure that tracked changes are fully resolved prior to submission.

7. PLOS authors have the option to publish the peer review history of their article (what does this mean?). If published, this will include your full peer review and any attached files.

Do you want your identity to be public for this peer review? For information about this choice, including consent withdrawal, please see our Privacy Policy.

Reviewer #2: No

Reviewer #3: No

Reviewer #4: Yes: Zaid Iyad Alkhatib

---

## [Decision Letter · Decision Letter 2]

4 May 2026

Phage Therapy for Recurrent Urinary Tract Infections: A Qualitative Study Using the Theoretical Framework of Acceptability

PONE-D-25-53420R2

Dear Dr. David Wenzel,

We’re pleased to inform you that your manuscript has been judged scientifically suitable for publication and will be formally accepted for publication once it meets all outstanding technical requirements.

Kind regards,

Awatif Abid Al-Judaibi, PhD

Academic Editor

PLOS One

Reviewers' comments:

Reviewer's Responses to Questions

**Comments to the Author**

1. If the authors have adequately addressed your comments raised in a previous round of review and you feel that this manuscript is now acceptable for publication, you may indicate that here to bypass the “Comments to the Author” section, enter your conflict of interest statement in the “Confidential to Editor” section, and submit your "Accept" recommendation.

Reviewer #1: All comments have been addressed

Reviewer #4: All comments have been addressed

2. Is the manuscript technically sound, and do the data support the conclusions?

Reviewer #1: Yes

Reviewer #4: Yes

3. Has the statistical analysis been performed appropriately and rigorously? 

Reviewer #1: Yes

Reviewer #4: Yes

4. Have the authors made all data underlying the findings in their manuscript fully available?

Reviewer #1: Yes

Reviewer #4: Yes

5. Is the manuscript presented in an intelligible fashion and written in standard English?

Reviewer #1: Yes

Reviewer #4: Yes

6. Review Comments to the Author

Reviewer #1: All comments have been addressed fully and carefully, with revisions made to improve clarity and quality.

Reviewer #4: (No Response)

7. PLOS authors have the option to publish the peer review history of their article (what does this mean?). If published, this will include your full peer review and any attached files.

Reviewer #1: **Yes:** Ali Bin Thani

Reviewer #4: **Yes:** Zaid Iyad Mohammad Alkhatib

---

## [Editor Report · Acceptance letter]

PONE-D-25-53420R2

PLOS One

Dear Dr. Wenzel,

I'm pleased to inform you that your manuscript has been deemed suitable for publication in PLOS One. Congratulations! Your manuscript is now being handed over to our production team.

Kind regards,

on behalf of

Professor Awatif Abid Al-Judaibi

Academic Editor

PLOS One